# A Pilot Study on the Feasibility of an Extended Suckling System for Pasture-Based Dairies

**DOI:** 10.3390/ani13162571

**Published:** 2023-08-09

**Authors:** Sandra Liliana Ospina Rios, Caroline Lee, Sarah J. Andrewartha, Megan Verdon

**Affiliations:** 1Tasmanian Institute of Agriculture, University of Tasmania, Burnie, TAS 7320, Australia; sarah.andrewartha@utas.edu.au (S.J.A.); megan.verdon@utas.edu.au (M.V.); 2Commonwealth Scientific and Industrial Research Organisation, Agriculture and Food, Armidale, NSW 2350, Australia; caroline.lee@csiro.au

**Keywords:** cow-calf contact, dam rearing, seasonal calving, pastoral systems, dam contact

## Abstract

**Simple Summary:**

There is an increased global effort to develop and identify the opportunities and challenges of extended dairy cow-calf suckling systems due to societal concerns about removing calves from cows soon after birth. Implementing a pasture-based system where the dams have half-day contact with their calves and are milked once a day could potentially serve as a viable choice for developing alternative practices in the dairy industry. However, most research on extended suckling systems has been conducted on indoor dairy farms. This paper describes an investigation into the productivity of cows and calves in a pasture-based extended suckling system featuring part-time cow-calf contact and once-a-day milking. Although suckled cows produced less saleable milk than commercially managed cows, their reduced milk yield did not persist after weaning. Dam-reared calves grew faster than commercially reared calves from weeks 3 to 9. The system offers a promising foundation for future large-scale and longitudinal research on the benefits, challenges, and scalability of pastoral cow-calf dairy systems. This approach would be in line with public expectations of enhanced animal welfare in the dairy industry by addressing the concern of removing calves from cows soon after birth.

**Abstract:**

This study investigated cow-calf productivity in a 10-week, pasture-based, extended suckling system featuring part-time cow-calf contact and once-a-day milking. A total of 30 dairy cows and their calves were assigned to two treatments: (1) cow and calf managed in an extended suckling system; or (2) cow and calf separated at birth and managed as usual. Cow-calf pairs grazed together during the day and spent the night separated by fence-line contact. The dams were reunited with the calves after once-a-day milking every morning. The commercial treatment pairs were separated after birth, and cows were milked twice a day and managed within the farm herd. Commercial calves were reared and managed as per commercial Australian practices. Cow-calf dams yielded 9 L/cow/day less saleable milk (*p* < 0.001), and their milk had lower fat (*p* = 0.04) but a higher protein percentage (*p* < 0.001) than commercial cows during pre-weaning. However, milk yield and composition were comparable post-weaning. Dam-suckled calves gained weight faster and were therefore weaned 2 weeks earlier than commercial calves, which were offered 8 L/day milk. This study has demonstrated a novel system of extended cow-calf suckling that could be practical to implement in pasture-based dairies. The long-term effects and scalability of the extended suckling system described here require further validation.

## 1. Introduction

Ninety-five percent of Australians view farm animal welfare as a concern, with 91% wanting reform to better protect farm animal welfare [1]. One area of community concern for dairy animal welfare relates to the separation of the calf from the dam [2]. Surveys from Canada, North America, Germany, and Brazil indicate that participants generally hold negative attitudes towards early separation in cow-calf management systems. They express ethical concerns over animal welfare, including the potential for pain and distress after separation [3,4,5]. Under typical commercial conditions, dairy calves are removed from their dam within 24 h of birth and reared artificially in groups of similarly aged animals. The separation of dairy calves from their dams soon after birth aims to reduce disease transmission, ensure calves consume enough high-quality colostrum, gain control of calf health and avoid cow-calf bonding that may lead to a future traumatic separation [6,7]. There is an increased global effort to develop and identify the opportunities and challenges of extended dairy cow-calf suckling systems [8,9,10,11,12,13,14] and support traditional dairy systems to meet changing societal demands in relation to farm animal management. 

The benefits of a cow-calf system could extend beyond meeting the changing expectations of society regarding dairy production. For example, early adopters cite improved animal welfare and personal ethical beliefs as benefits after transitioning to dairy systems which incorporate some level of cow-calf contact [15,16]. Most dairy farmers, however, maintain concerns over the practicality of cow-calf systems and their effects on labour and animal welfare [16]. Extended suckling systems need to address these concerns if they are to be considered a viable alternative system for the majority. 

Research from indoor dairy systems shows that dam-suckled calves may have better welfare and performance, as shown by lower serum cortisol during colostrum feeding, enhanced cognitive and social skills, improved growth, reduced fearfulness of novel objects, and fewer abnormal behaviours, such as cross-suckling, compared to calves that are separated from their dam at birth [6]. Extended suckling systems may also have positive effects on cow health, e.g., a systematic review by Beaver et al. [17] concluded that suckling systems could decrease the risk of mastitis.

There are challenges to extended suckling systems, however, including a reduced quantity of milk available for sale during the suckling period, impaired milk letdown, severing the dam-calf bond at a later age, and weaning calves off high milk volumes [6,17]. Emerging evidence of reduced milk fat and increased milk protein during the suckling period also requires further exploration [8,18]. 

Most research on extended suckling systems has been conducted on indoor dairy farms [6,17], with smaller herd sizes than those found in pasture-based dairies and where year-round calving means there are fewer calves to manage at any one time [7]. Other challenges to extended suckling systems in pastoral dairies include the housing of young calves outdoors during the winter, the frequent moving of cows within and between paddocks, and increasingly large distances walked by cows to and from the dairy. The low-input, low-output operation of pastoral dairy farms may also mean a high percentage of saleable milk is sacrificed to the calf during the suckling period compared with indoor dairy production systems. For example, a recent study on six male calves housed with full-time contact with their dams at pasture in a twice-a-day milking system found that calves consumed nearly 20 kg of milk per day by weaning at 100 days of age [10]. This equalled a 55% reduction in total daily milk yield around weaning. 

An extended suckling system designed specifically for pastoral dairies is required for pasture-based management systems to capitalise on the benefits of extended suckling while continuing to be practical and profitable. As reviewed by Verdon [7], half-day cow-calf contact with once-a-day milking may be a more practical alternative to full-time suckling in pasture-based dairies. This system could also address some of the dairy industry’s challenges, such as labour shortages, particularly around calving [19]. Cows separated from their calves overnight also produce more milk in the morning than cows with full-time contact with their offspring and are better able to maintain their body condition across early lactation [18]. 

This paper describes an investigation into the productivity of cows and calves in a pasture-based extended suckling system featuring part-time cow-calf contact and once-a-day milking. It is hypothesised that calves reared by their dams will grow faster but have a larger growth check post-weaning. A secondary hypothesis is that cows nursing their calves will have improved body condition and will produce less saleable milk while nursing but comparable milk volumes to commercial cows post-weaning. 

## 2. Materials and Methods

### 2.1. Ethical Statement and Animal Welfare

All animal procedures were conducted with prior institutional animal ethics approval (University of Tasmania Animal Ethics Committee A0024805) under the requirements of the Tasmanian Animal Welfare Act (1993) and in accordance with the National Health and Medical Research Council/Commonwealth Scientific and Industrial Research Organisation/Australian Animal Commission Code of Practice for the Care and Use of Animals for Scientific Purposes.

### 2.2. Animals and Experimental Design

The research was performed at the Tasmanian Institute of Agriculture (TIA) Dairy Research Facility (TDRF) at Elliot (41°08′ S, 145°77′ E; 155.0 m a.m.s.l.) in North West Tasmania, Australia. The mean monthly rainfall from July to September 2021 was 114 ± 35.9 mL. The mean daily temperature ranged from 5.7 °C to 13.8 °C, with a lowest temperature of −1.2 °C and a highest temperature of 18.2 °C [20,21]. 

Here, 34 pregnant dairy cows (*Bos taurus*) were selected from the TDRF’s herd of cows that were confirmed pregnant with sexed semen for female calves (~20 weeks pregnant). Cows were assigned to one of two treatments balancing parity (mean 2.2 ± 1.3), body condition (4.2 ± 0.2), and live weight (511 ± 55 kg). The treatments were: (1) cow and calf managed in TIA’s extended suckling system (also called “cow-calf suckling”); or (2) cow and calf separated at birth and managed as usual (commercial). Fou of the 34 selected cows were excluded after birthing a male calf. The remaining 30 cows and their dairy heifer calves were included in this study. Sixteen cow-calf pairs (11 Fresian and 5 Fresian × Jersey cows) were allocated to the extended suckling system. The 14 commercially managed cows (7 Fresian and 7 Fresian × Jersey cows) and calves were separated within 6 h of birth and managed as per normal practice (details of management provided later). 

Two calves assigned to the extended suckling system were born at the start of the calving period and during remarkable adverse weather conditions (i.e., they were small and temperatures were under 2 degrees, with heavy rain and/or extreme wind). These calves were instead allocated to the control group on the basis that we could better ensure their welfare, considering how little is known about the health of calves born into a pasture-based extended suckling system during the winter. Their dams also entered the control treatment. These two cow-calf pairs were replaced with cows (and their calves) of equal parity and live weight that were originally assigned to the control treatment. 

### 2.3. Husbandry Practices

Disbudding of 12 cow-calf suckling and 10 commercial calves (4 suckling and 4 commercial calves were polled) was performed at 6 weeks of age by a veterinary surgeon. The calves were kept in their home pen and sedated (xylazine hydrochloride, 0.05 mg/kg IM). Local anaesthesia (lignocaine hydrochloride 2%, 2 mL per each horn bud, SC) was administered at the base of the horn buds and a non-steroidal anti-inflammatory drug (meloxicam, 1.5 mg/kg, SC) was administered for pain relief.

All calves were weaned from milk after their group reached an average live weight of 90 kg. 

### 2.4. Calving and Post-Calving Management 

Pregnant cows were moved to a calving paddock (~40 × 90 m) that contained a semi-enclosed shed for shelter one week prior to their expected calving date. Calving cows were checked at least four times per day: before morning milking (5:00 AM), morning (10:00–11:00 AM), afternoon milking (4:00 PM), and evening (8:00–10:00 PM), with occasional in-between monitoring to assess calving progress. All calves were born over a 2-week period, and there were no calving difficulties.

#### Management of the New-Born Calves

Calves were collected as soon as possible after birth (within 5–6 h). Their navels were sprayed with a 7% iodine solution, and identification ear tags were fitted at the paddock. They were then gently transported in a calf trailer. The 14 commercial calves were taken to their ‘group pen’ as per commercial practice. Their dams entered the colostrum herd. The 16 suckling calves were taken to a ‘calving shed’ (next to the calving paddock) with their dam. 

On arrival at the pen/shed, calves were weighed, their navels were again sprayed with iodine solution, and they were fed 2 L of quality colostrum (Brix refractometer score > 23). Another 2 L of colostrum was fed within 6 h. The suckled calves were briefly separated from the dam (~20 min) during these management procedures, after which cow and calf remained together until the next scheduled milking. Once milked, suckling cows were moved (with their calves) to the extended suckling paddocks, which were located about 300 m away from the milking shed (Figure 1). Individual cow-calf pairs were held for 30 min in a ‘bonding’ pen (4.2 m × 4.2 m) to ensure dam-offspring recognition before they entered the extended suckling system.

### 2.5. Commercial Treatment

Commercial calves were hand-reared in a 43 m^2^ pen of 14 animals (stocking density 3.04 m^2^/calf) within a semi-enclosed barn with natural light and ventilation. They were offered 8 L of whole milk per day, water nipples, and milled grain ad libitum via a robotic feeder (DeLaval automatic calf feeder CF 150). These robotic feeders integrate a milk and grain station, provided with a rubber teat and grain trough. The feeders gradually distributed the milk over a 24-h period with a maximum of 3 L able to be consumed in a single suckling session. This allowed the calf to have access to the full amount of milk per day without being able to consume all 8 L at once. In line with commercial practice in Australia, roughage was not provided to these calves. The pen floor was covered with woodchip bedding. The concrete floor surrounding the robotic feeder was hosed out daily. The feeder lines were flushed, and their teats were cleaned and disinfected daily. A 25-cm heavy-gauge vinyl jump ball (Kmart, SKU: P42187110, Sydney, Australia) was provided with a grab handle and hung from the shed ceiling in the middle of the pen, and a 25 cm × 10 cm brush was fitted to the pen wall for enrichment. 

Commercial cows were managed within the commercial herd of 300 animals and milked twice per day, as per commercial practice. They were fed 21.4 kg DM/day/cow by grazing perennial ryegrass (*Lolium perenne* L.) with a daily fresh allocation of pasture (~15.5 kg DM) that was supplemented with silage (3–6 kg DM) fed in the paddock and ~5 kg of concentrate fed during milking (split over morning and evening milking events).

### 2.6. Extended Suckling System

#### 2.6.1. Design of the Extended Suckling System

Cow-calf pairs in the extended suckling treatment were housed outdoors in pastured paddocks (Figure 1, Appendix A). Four paddocks were used, providing a total of 4 hectares for rotational grazing. The paddocks were divided longitudinally into AM and PM allocations using four-strand electric fencing. For the first week of the study, a one-metre strip of tall grass was made accessible only to calves at the front of the AM allocation. This ‘nap strip’ capitalised on the calves’ natural inclination to rest in tall grass in the first week of birth, while the dam grazed nearby [22,23], and provided some protection from the weather. Calves were fitted with waterproof jackets during the first 2 weeks of rearing. A calf pen was constructed at the end of the paddock (pen size average = 361 m^2^) using non-electrified wallaby-proof fencing. The pen had two shelters (2 m × 1.5 m and 1.2 m in height) that could accommodate up to 10 calves each. The opening of the calf pen included a “double-gate”, which prevented cow access to the calf pen while permitting free movement of calves.

#### 2.6.2. Management of the Extended Suckling System

Suckling calves had access to their dams during the day, where they suckled ad libitum. The cow-calf pairs were separated overnight (approximately 17:00–08:00 h) during which the calves were locked in the calf pen. Fence-line contact between cows and calves was maintained during separation. There was no chance of suckling occurring through the fence (~4 cm mesh spacings). Calves had ad libitum access to water and pasture when with the cows and when separated. Grain was also offered ad libitum inside the calf pen. Suckled cows were milked once a day in the morning before being reunited with their calves during daylight hours. Anecdotal observations suggested that the teats of some cows were becoming damaged by vigorous calf suckling after being re-united, presumably because the udder was dry immediately after milking. The cows’ discomfort has been previously reported and attributed to an empty udder after milking [24]. To address this, the dams remained alone in the paddock for 30 min before the calves were let out of their pen. Cows were offered the same amount of feed as the commercially managed cows, i.e., 21.4 kg DM/day/cow consisting of grazed perennial ryegrass pasture supplemented with silage (~5 kg/day/cow provided in the paddock for the AM feed and in a trough for the PM feed) and concentrate pellets (~5 kg/day/cow, half provided in the dairy during the morning milking and the other half sprinkled over the silage in the trough for the PM feed). 

#### 2.6.3. Management of the Evening Cow-Calf Separation

A fresh allocation of pasture with silage supplementation was made available to the cows at separation. This enticed the cows to move through the ‘separation gate’ from the day paddock to the night paddock. The cows’ afternoon allocation of concentrate was distributed in long troughs in the night paddock to further encourage movement from the day paddock. Cow-calf pairs were gently herded from the day paddock and walked to the separation gate (see Figure 1). The cows waited at the gate while the calves were herded to their pen. The separation gate was opened once all the calves had passed under the double gate into their calf pen and were shut away. Up to 3 handlers performed the separation during the first 7 days, after which it was managed by a single handler within 10 min. The separation of cows and calves overnight was performed to simplify the collection of cows for the morning milking, optimise udder fill before the morning milking, and accustom calves to handling and separation. No cows in this study became visually agitated during separation. Some calves were reluctant to enter the pen during early training sessions but were able to be calmly and gently pushed inside. Calves mostly entered the pen independently after adjusting to the system.

### 2.7. Weaning

Calves were weaned as a group once the group reached an average live weight of 90 kg. Commercial calves were weaned following a gradual reduction in milk from 8 L to 4 L over 14 days. 

A two-stage weaning method was used for suckling calves to break their nutritional dependence on the dam before physical separation. Calves were fitted with a plastic anti-suckling device that prevented the calf from suckling the dam but allowed them to graze, consume grain, and drink water (QuietWean nose-flaps, USA; www.quietwean.com accessed on 10 May 2021). The cows were milked twice a day after the nose flaps were fitted. The cow-calf pairs were separated three days later. On the day of separation, cows were moved to a different paddock away from their calves after their morning milking. The calves were then moved on a calf transport trailer to a paddock approximately 1 km away. The cows remained separated from the main milking herd for 5 days after weaning. This allowed close observation as they adjusted to the separation. After this period, the suckled cows re-joined the milking herd and continued their lactation under commercial farm management. Calves from both treatments were mixed post-weaning and continued to be reared as a single cohort under commercial conditions.

### 2.8. Variables Recorded

#### 2.8.1. Milk Yield

Individual cow milk production (L) was recorded using a DeLaval Alpro milk metering system (DeLaval International AB, Tumba, Sweden). The milk production of commercial cows was recorded during morning and afternoon milking. These were summed to calculate their daily milk production. Milk production of suckled cows was recorded during their only milking, which was in the morning. Following weaning, when all cows were managed into the commercial herd, the daily morning and afternoon milk production were recorded and summed for all cows for the remainder of the lactation. 

For analysis, the stages of lactation were categorised as pre-weaning (weeks 1–10), mid-lactation (weeks 11–28), and late lactation (weeks 29–44). The average milk yield (litres/day/cow) was calculated for each week and then averaged per lactation stage for the analysis. The daily milk yield of cows was also summed across the full lactation to determine the total lactation milk per cow (litres/lactation/cow).

#### 2.8.2. Milk Composition

Milk samples were collected from individual cows every second month throughout the full lactation during the morning milking (*n* = 6 samples/cow). Samples were analysed by TasHerd using a Bentley B2000 infrared milk analyser (Bentley Instruments Inc., Chaska, MN, USA). Somatic cell count is expressed as the number of cells per microlitre (µL) of milk, and protein and fat concentrations are expressed as a percentage. 

For the analysis, cows with cell counts of ≤ 200 cells/µL were classified as at ‘low risk’ of mastitis, and cell counts ≥ 200 were considered as ‘at risk’ of mastitis [25]. For analysis purposes, the last test performed in each lactation stage was considered (*n* = 3 samples/cow).

The fat and protein corrected milk yield (FPCM) was calculated for each sample date using Equation (1) Sjaunja Sjaunja [26]:FPCM = milk yield (kg.) × ((0.383 × % fat + 0.242 × % protein + 0.7832)/3.140)(1)

The FPCM provides an indication of the amount of milk produced adjusted to a standard of 4.00% fat and 3.40% total protein. 

#### 2.8.3. Cow Live Weight and Body Condition

Cow live weight and body condition score (BCS) were measured daily using walk-over scales and a 3D BCS camera as the animals exited the dairy through a raceway (DeLaval AWS100 automatic weighing system and DeLaval BCS Automatic Body Condition Scoring system, Tumba, Sweden). The AWS100 software uses an algorithm that discards weights that differ greatly from the 7-day average weight of individual animals (e.g., instances where more than one cow is on the platform). Thus, the 7-day average weight data can provide the most accurate representation of cow live weight. Body condition was scored using the New Zealand 10-point scale [27]. Daily BCS were recorded and then averaged per cow and week daily using the DeLaval BCS automated scoring system, and then the weekly mean was calculated. 

#### 2.8.4. Calf Live Weight

Calves were individually weighed on a portable stock scale in the calving shed at birth (week 0) and on a purpose-built platform fitted within a calf-weighing chute set in the home pen at weeks 3, 6, and 9, (Figure 1). The accuracy of the portable scale was assessed before and during each weighing event by measuring a known weight. Suckling calves were weighed in the morning before joining their dams (~8:00 AM). Commercial calves were weighed after the suckling calves (~9:00 AM). 

Average daily gains (ADGs) were calculated from birth until week 3, from birth until week 9, and from week 3 to week 9. The difference in weights between the two-time points was calculated and divided by the number of days between them to determine the ADG.

#### 2.8.5. Calf Milk and Grain Intake

Individual milk and grain consumption of the commercially managed calves was recorded daily via the calf auto feeder. The grain intake of suckled calves was determined at the group level. A known quantity of grain was provided each morning (enough to ensure ad libitum intake), and the amount refused was measured 24 h later. The estimated milk intake of suckled calves at weaning was calculated by subtracting the suckled cows’ average milk yield (litres/cow/day) during the last week pre-weaning from their average milk yield in the first week post-weaning [28].

#### 2.8.6. Cow Behaviour

The transition of the suckled cows into the commercial herd was studied by measuring cow behaviour prior to (weeks 6–9) and after (weeks 11–14) the transition into the commercial herd. Each cow in both treatment groups was fitted with a neck-collar-mounted activity monitoring system (MooMonitor^+^, Dairymaster Inc., Kerney, Ireland). The commercially available MooMonitor^+^ uses a 3-dimensional accelerometer to determine cow movement and head direction and, based on proprietary algorithms, distinguishes between ruminating, resting, and feeding behaviour. Using the MooMonitor^+^, the present study continuously monitored the time (hours) cows spent ruminating, resting, and feeding. These data were used to calculate cow daily time budgets during the 3 weeks prior to weaning (weeks 6–9) and 3 weeks post-weaning (weeks 11–14) once the cows had been reintroduced into the commercial herd. 

### 2.9. Statistical Analysis

#### 2.9.1. Analysis of Cow Data

Two cows did not complete a full lactation (one commercial cow was dried early due to low milk yield, and one suckled cow was culled due to cancer). Statistical analysis was performed using SPSS software (SPSS 28.0.1.0; SPSS Inc., Chicago, IL, USA). Data collected during the pre-weaning period of this study were included in the model, but post-weaning lactation milk production data were removed prior to statistical analysis. Data were assessed for normality using visual means (histograms and Q-Q plots) supported by statistical tests of normality (Kolmogorov–Smirnov statistic). Body condition score and milk fat and protein percentages were logarithmically transformed prior to analysis with a parametric statistical test. Unless otherwise stated, means and standard deviations are reported in the results. Statistical significance was assumed at *p* ≤ 0.05.

The effects of treatment, lactation stage (pre-weaning, mid-lactation, late-lactation), and their interaction on average daily milk yield, cow live weight and body condition, percentage fat and protein, and cow behaviour were analysed using linear mixed models (LMM). The data were treated as repeated measures made on the same experimental units with an unstructured covariance structure. The effects of the treatment on total milk yield over the full lactation and on the FPCM at each sampling day were assessed using an independent-sample *t*-test. A chi-square test for independence (with Yates’ continuity correction) was used to assess treatment effects on the proportion of cows’ ‘at risk’ of mastitis (cell counts of ≤200 cells/µL) using data collected at the end of each lactation stage. 

#### 2.9.2. Analysis of Calf Data

A general linear mixed model was used to assess the effects of treatment, week (i.e., birth weight, weeks 3, 6, and 9 of age), and their interaction on calf weight. Repeated observations of the calf over weeks were included in the model as a random effect with an auto-regressive covariance structure. Calf average daily weight gains from weeks 0 to 3, 0 to 9, and 3 to 9 were analysed using a linear mixed model. Calf birth weight was included in the model as a random effect.

Data on the grain intake of suckled calves was taken at the group level and could not be analysed. These data were subject to descriptive analysis.

## 3. Results

### 3.1. Milk Yield

There was a significant interactive effect of the management system and lactation stage on daily milk yield (F_1,28_ = 113, *p* < 0.001; Figure 2). Suckled cows produced approximately 9 L less milk per day than commercial cows while with their calves (16.7 ± 3.9 vs. 25.9 ± 3.0 L, F_1,28_ = 69.3, *p* < 0.001), but milk production was comparable between the treatments post-weaning (mid-lactation 24.2 ± 1.6 in suckled cows vs. 23.3 ± 2.6 L in commercial cows, F_1,28_ = 2.5, *p* = 0.137; late-lactation 18.9 ± 1.8 vs. 17.6 ± 2.3 L, respectively, F_1,28_ = 0.86, *p* = 0.36). The total milk produced by suckled cows across the lactation was 8% less than that produced by the commercial group, corresponding to the 9 L/day/cow lost during the suckling period (6213 ± 495 vs. 6731 ± 557 L, respectively; *t* (26) = 2.60, *p* = 0.015, two-tailed).

### 3.2. Milk Composition

Suckling did not increase the risk of mastitis in the pre-weaning, mid-lactation, or late-lactation periods (Table 1).

The milk of commercially managed cows had a higher fat but lower protein percentage than the milk of the suckled cows during the pre-weaning stage of lactation (fat: F_1,28_ = 4.6, *p* = 0.04; protein: F_1,28_ = 69.9, *p* < 0.001). Consequently, suckled cows had 32% less FPCM yield than commercial cows during pre-weaning (*t* (26) = 3.43, *p* < 0.001). The treatment did not affect milk composition during mid- and late-lactation (Table 2).

### 3.3. Cow BCS and Weights

There were no effects of treatment or a treatment × lactation stage interaction on cow live weight or BCS (Table 1), but live weight and BCS increased over the lactation in both treatment groups (live weight F_2,26_ = 131.09, *p* < 0.001; BCS F_2,28_ = 29.69, *p* < 0.001; Table 1). 

### 3.4. Calf Weights

Calf weight was affected by a treatment × week interaction (F_3,28_ = 20.0, *p* < 0.001). Calves in the suckling system were by chance 5 kg heavier at birth than commercial calves (40.0 ± 4.18 vs. 34.8 ± 5.43 kg). Three small calves were born into the commercial group (birth weight 27.2 ± 5.01 kg vs. 36.5 ± 3.28 kg for the rest of the commercial group). This includes two calves who were reallocated from suckling to commercial treatment after being born at the start of the calving season under exceptionally adverse climatic conditions. Weights did not differ between the treatments at weeks 3, 6, or 9 (Figure 3).

Calf growth from weeks 0 to 9 was not affected by treatment (ADG 807 ± 107 commercial vs. 827 ± 84 g/day suckling, F_1,28_ = 0.36, *p* = 0.56). Commercial calves grew faster than suckling calves from weeks 0 to 3 (ADG 839 ± 162 commercial vs. 697 ± 108 g/day suckling F_1,28_ = 4.67, *p* = 0.04), but suckling calves grew faster than commercial calves between weeks 3 and 9 (ADG 906 ± 91 vs. 785 ± 103 g/day; F_1,28_ = 11.73, *p* = 0.002). Suckling calves were consequently weaned two weeks earlier than commercial calves (9 ± 0.4 versus 11 ± 0.6 weeks of age). 

### 3.5. Calf Milk and Grain Intake

Commercial calves consumed an average of 6.4 ± 0.52 L of the allocated 8 L whole milk/day (80% of allocation). Milk consumption increased from weeks 1 to 3 as the calves trained to the auto-feeder and decreased from week 9 due to the gradual weaning. Excluding these weeks did not substantially change the average milk consumption of commercial calves (6.8 ± 0.40 L from weeks 3 to 9). By contrast, suckling calves were consuming an estimated 9.9 L milk/day by weaning.

Total grain intake for commercial calves during the 10-week rearing was 18.4 ± 0.24 kg/calf compared with 2.7 ± 2.6 kg/calf for suckling calves. The grain intake of suckling calves increased by 33% in the 3 days after the quiet-wean nose-flap was fitted, meaning that daily grain intake at weaning was similar between the treatments (675 g vs. 643 ± 0.33 g, respectively). 

### 3.6. Cow Behaviour

Pre-weaning, cows with calves that were milked once a day spent 22 min less ruminating, 40 min more resting, and 110 min more feeding than cows managed in the commercial herd and milked twice a day (F_1,28_ = 5.05, *p* = 0.03; F_1,28_ = 8.22, *p* = 0.008 F_1,28_ = 39.5, *p* < 0.001). The behaviour of suckled and commercially managed cows was comparable post-weaning (Table 3).

### 3.7. Health

There were no calf health issues (i.e., respiratory disease, diarrhoea, anorexia) requiring veterinary assistance during the pre-weaning period. Seven suckling calves and four commercial calves presented with mild scours. They were provided electrolytes, mostly as a preventative measure, but two commercial calves also required antibiotic treatment. A negative parasitology egg count test was obtained from the suckling and commercial calves two weeks after weaning. 

Three cows from each group required antibiotic or/and anti-inflammatory treatment for mastitis, and one cow from the suckling group was treated for recurring lameness after a veterinary assessment. One suckled cow was involuntarily culled during mid-lactation due to cancer. The remaining cows all successfully became pregnant for their next gestation.

## 4. Discussion

This pilot is the most comprehensive study of a potential extended cow-calf suckling system for large pasture-based dairies in the published literature. Of the 70 peer-reviewed studies examined by Beaver, Meagher, von Keyserlingk, and Weary Beaver, Meagher, von Keyserlingk, and Weary [17], only two were conducted in Australia and one in New Zealand, and these focused on Johne’s disease or udder health between separated or suckled cows. The nature of large, seasonal-calving, pasture-based dairy systems has been a barrier to cow-calf suckling systems [7]. For example, New Zealand dairy farmers identified increased labour, infrastructure, and herd management as key barriers preventing extended suckling [16]. This study has demonstrated a system of extended cow-calf suckling that could be practical to implement in pasture-based dairies. It offers a promising foundation for future large-scale and longitudinal research on the benefits, challenges, and scalability of pastoral cow-calf dairy systems.

### 4.1. Milk Production and Composition

Consistent with Barth [8], who studied a barn-based cow-calf system, suckled cows in this study produced less milk across their total lactation than commercially managed cows. Barth [8] attributed this to the reduced saleable milk collected from suckled cows during the pre-weaning period. In this study, suckled calves consumed an estimated 9.9 L of milk/day at 10 weeks of age, compared with 6.4 L/day consumed by commercial calves. Besides that, they consumed fewer grains than commercial calves. In a pasture-based extended suckling system of four cow-calf pairs with full-time contact and twice-a-day milking, Mac, Lomax, and Clark [10] found that dairy bull calves were consuming an estimated 20 L/day by weaning. The difference in estimated drinking volumes of calves in the present study to those in the study by Mac, Lomax, and Clark [10] is likely a combination of the different grain fed and sexes studied, the greater access of calves to cows, and the later weaning in the latter study (weaning at 15 weeks). 

In accordance with previous studies, we saw no effects of cow-calf suckling on post-weaning milk yield [6]. While saleable milk production was lower during the pre-weaning period, this may be partially compensated by earlier weaning of the suckled calves, reduced amount of grain for calf feeding, and reduced labour due to once-a-day milking. It is unclear whether cow-calf systems will reduce labour associated with calf rearing, but it is likely the nature of rearing responsibilities will change. For example, routine tasks such as carting milk to the calf shed and feeding out calves or hosing pens will be eliminated, but health assessments and early identification of poor health may become more vital. As noted in the reviews of Verdon [7] and Meagher, Beaver, Weary, and von Keyserlingk [6], economising extended suckling systems based on milk sales alone provides a reductive comparison to conventional dairy systems. Therefore, a comprehensive analysis across the entire system (i.e., heifer growth, first lactation milk yield, and cow health) is necessary to evaluate the costs and benefits of pasture-based cow-calf suckling systems. 

Suckled cows produced milk with a higher percentage of protein, but lower fat and FPCM, during the preweaning stage of lactation than commercial cows. Barth [8] and Nicolao et al. [11] reported similar results in a housed cow-calf system, but Wenker, Verwer, Bokkers, Te Beest, Gort, De Oliveira, Koets, Bruckmaier, Gross, and Van Reenen Wenker, Verwer, Bokkers, Te Beest, Gort, De Oliveira, Koets, Bruckmaier, Gross, and Van Reenen [14] did not find differences in mean protein. These differences may be attributed to variances in genetics (Holstein and Red Pied vs. Holstein × Friesian cows) and the frequency of milk sampling (monthly vs. triweekly). Milk fat and protein content are inversely proportional (i.e., the higher the fat, the lower the protein content), and changes in one of these components may indicate changes in the other [29]. It is unlikely that the differences in milk protein in the present study were dietary related. Commercial and suckled cows were offered a comparable diet with the same dry matter content. Other research from housed management systems similarly suggests comparable metabolic performance between suckled and commercially managed cows [14]. Alternatively, the frequent udder emptying from nursing the calf may reduce fat milk content and, in turn, could have led to an increase in milk protein percentage. Nicolao et al. [11] found that the timing of suckling affects milk composition, with lower milk fat when calves suckle after or between milkings and higher milk fat when they suckle before a milking. As discussed by Nicolao et al. [11], when suckling occurs after milking, the calves mainly consume residual milk that is higher in fat content. By contrast, when they suckle before milking, they are mainly consuming cisternal milk, which has a lower fat content (discussed by Nicolao et al. [11]). Future research should further explore the relationship between suckling and reduced fat content in milk, especially considering that dairy farmers are often paid per kg of fat and protein [30].

### 4.2. Mastitis

The risk of mastitis (determined by SCC) did not differ between suckling and commercial treatments across the lactation. This adds to the scientific evidence reviewed by Beaver, Meagher, von Keyserlingk, and Weary Beaver, Meagher, von Keyserlingk and Weary [17], which suggests extended suckling doesn’t increase the risk of intramammary disease. Calf saliva and frequent suckling may remove bacteria from the teat skin to reduce the risk of infection [31]. Frequent suckling may also promote improvements in mammary tissue and milk yield through the release of stimulating hormones such as oxytocin, prolactin, and growth hormone [32]. The results of this study, as well as those of other research [17], indicate that concerns about mastitis should not prevent growers from implementing extended suckling systems.

### 4.3. Cow Behaviour

In the 3 weeks prior to weaning, suckled cows milked once a day spent 26% more time feeding, 14% more time resting, and 4% less time ruminating than commercial cows milked twice a day. These results are consistent with once-a-day milked cows spending less time walking to the dairy and ruminating while waiting to be milked, meaning they have more time to rest and feed in the paddock [33,34]. These results may have been exacerbated by the fact that cows in the commercial group were managed in a larger herd of 300 animals compared to the herd of 16 cows in the suckled group, extending waiting times to be milked and meaning that the paddocks they grazed could be at longer distances from the dairy compared to those in the extended suckling system. 

There were no significant differences in the behaviour of suckled and commercially managed cows once the dams were separated from the calves and re-joined the commercial herd. These results support the hypothesis that suckled cows adjust quickly to commercial herd management post-separation from the calves. Cow separation distress at weaning is among the main concerns of farmers regarding extended suckling systems [16]. Delayed separation increases cow and calf vocalisations and activity post-separation at 1 day vs. 14 days of separating the calves from their dams [35]. The calves in our pilot extended suckling system experienced overnight separation from the dam from birth, which may aid in the transition to weaning. Our system also included a 2-step weaning process to break the nutritional bond before the physical separation. While we did not take records on distress around weaning and separation in this study, other research suggests that the stress of suckled calves at separation from the dam is lower when they are managed in a partial cow-calf contact system compared to calves with full-contact [36]. Half-day contact systems might help prepare cows and calves for separation and weaning. Future research featuring our cow-calf management system could focus on the behavioural and physiological responses of cows and calves to weaning and separation in order to fully elucidate the extent and duration of stress during this period.

### 4.4. Calf Growth

Commercial calves were by chance 5 kg lighter than suckled calves at birth but gained more weight than suckled calves in the first 3 weeks of life, so live weights were similar by week 3. Suckling calves had access to shelters during the day and night but would typically spend daylight hours in the paddock with their dam, where they were exposed to environmental extremities. By contrast, commercial calves always remained indoors, sheltered from wind and rain. The temperature outdoors dropped as low as 0.1 degrees Celsius during the first weeks of this study [21]. The ADG of indoor-reared calves has been shown to reduce by 12 g for each 1 °C decrease in temperature [37]. The suckling calves in this study may have spent additional energy on thermoregulation than the commercial calves, potentially reducing the energy available for growth in the first few weeks of life while the suckling routine was established. A second, but perhaps a less likely explanation, is due to the housing of suckled calves with increased environmental enrichment and space allowance, resulting in increased activity and hence energy expenditure. Indeed, cow-contact calves have shown increased locomotory play due to a higher common space allowance than calves housed indoors in groups [38]. However, calves typically ‘hide’ and have reduced mobility in the first weeks of life [22,23]. 

Once suckling was established, however, the ADG from weeks 3 to 9 was higher for suckled calves, and they were weaned one week earlier than commercially reared calves. This is despite separated calves being offered a higher milk allowance in this study compared to that typically provided to commercially managed calves (4–5 L/day; [37]). The estimated higher milk allowance of suckled calves (9.9 L/day) in this study aligns with that reported in other half-day contact and restricted suckling systems [38,39] and is likely implicated in their higher weight gain. Other research also finds the growth of restricted suckled calves is similar to calves in full-contact cow-calf systems [39,40]. Thus, half-day contact systems may offer a compromise to full-time contact systems by minimising the reduction in milk production while continuing to accelerate calf growth. The availability of pasture and silage that was offered to the dams of suckled calves may also be implicated in their accelerated ADG from weeks 3 to 9. Anecdotally, the calves in this study were seen grazing by the side of their dams from week 1. Incorporation of forage in the diet of suckling calves from an early age may have facilitated rumen development. For instance, Mac, Lomax, and Clark [10] observed the ruminal development of grazing dam-reared Holstein male calves with low grain consumption (<1 kg/calf/d) to be comparable to conventionally reared calves. The inclusion of non-structural carbohydrates derived from grains is traditionally encouraged to promote the development of rumen [41]. However, emerging research demonstrates that the consumption of fibre from three days of age facilitates the initial establishment of rumen microbiota and intestinal bacteria in dairy calves [42]. A future study including faecal microbial analysis is suggested as a non-invasive method to determine calf rumen development in pastoral cow-calf suckling systems.

It is important to note that the weight of the suckling calves was measured before rejoining their dam, meaning that they had not received milk since the previous evening. On the other hand, the commercial calves had access to milk via an auto-feeder during the night. It is unclear whether differences in stomach content affected the live weights of calves. However, the growth differences between treatments may be greater than those reported here.

## 5. Conclusions

This pilot study describes a cow-calf rearing system for pasture-based dairies based upon once-a-day milking and half-day contact as an alternative to separation from the dam at birth. Our cow-calf rearing system was practical to manage, and there were no negative effects on cow or calf welfare. It provides a foundational system using which further studies could explore the long-term effects, economics, and scalability of cow-calf suckling in pastoral dairy settings.

## Figures and Tables

**Figure 1 animals-13-02571-f001:**
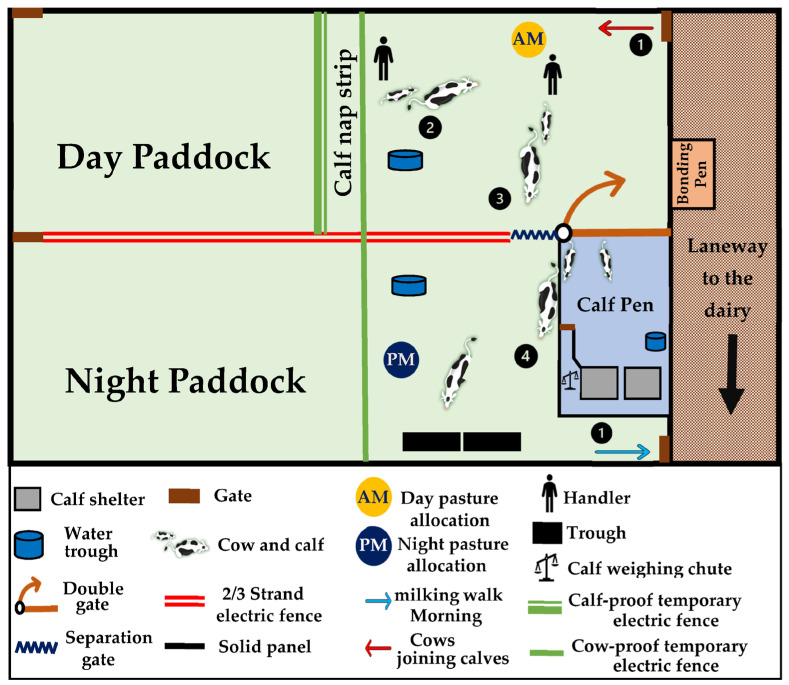
The TIA cow-calf suckling system. Schematic (not to scale) of the paddocks used to house the cow-calf pairs. Daily management steps are represented by numbers 1–4 on the figure and are described in the following sentences. (1) Milking: In the morning, after spending the night separated from the calves, cows were walked from the experimental paddock to the dairy shed for milking. (2) Cow-calf contact: Dams were returned to the day paddock after milking, and calves joined them 30 min later. Cows and calves spent the day in the AM area, with full contact. (3) Separation: Cows and calves were separated each night at approximately 1700 h. Cow-calf pairs were first walked to the gate separating the day and night paddocks. The calves were drafted into their pen while the cows waited at the gate. The separation gate was opened after all the calves were in the pen and locked away for the night, giving the cows access to the night paddock. (4) Overnight fence contact: The cows spent the night in the PM area, and the calves were within their pen. Fence-line contact was maintained.

**Figure 2 animals-13-02571-f002:**
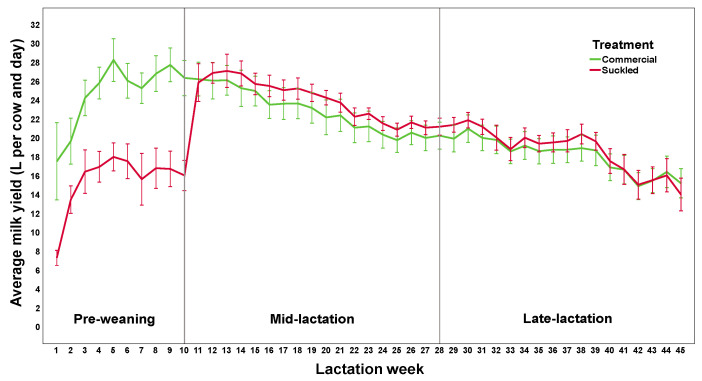
The mean daily milk yield (litres) produced per cow per week of lactation in the commercial and suckled treatments. Error bars: 95% CI.

**Figure 3 animals-13-02571-f003:**
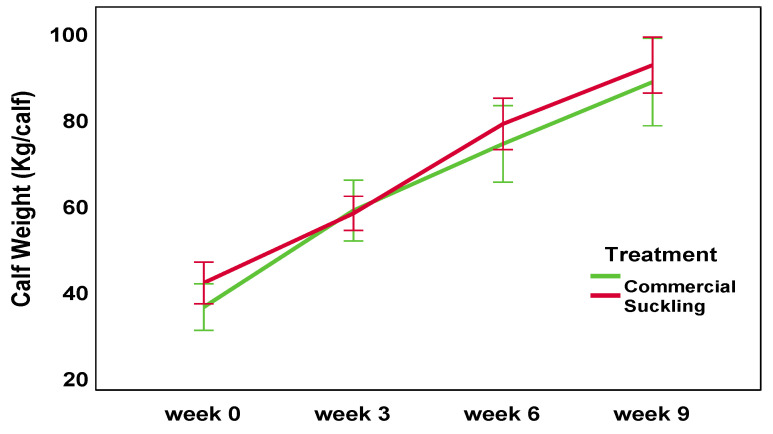
The mean calf weights (kg) at birth and weeks 3, 6, and 9 of the rearing period. Average daily gain (ADG) was significantly different between weeks 3 and 9 of rearing (*p* ≤ 0.05). Error bars: 95% CI.

**Table 1 animals-13-02571-t001:** Mastitis risk, live weight (LW) and body condition score (BCS) of cows separated from their calves (commercial) or in an extended cow-calf suckling system (suckled) over three stages of a single lactation. Percentage of cows with SCC > 200,000 (‘mastitis risk’) or mean values ± standard deviations (LW and BCS) are presented.

Variable	Lactation Stage	Treatment		Test Statistic ^1^	*p*-Value
		Commercial	Suckled		
Mastitis risk	Pre-weaning	15.4%	6.7%	X^2^_1,28_ = 0.02	0.90
	Mid-lactation	23.1%	6.7%	X^2^_1,28_ = 0.49	0.49
	Late-lactation	23.1%	13.3%	X^2^_1,28_ = 0.03	0.86
LW (kg)	Pre-weaning	512 ± 46.3	535 ± 55.5	F_1,28_ = 1.44	0.24
	Mid-lactation	544 ± 38.9	575 ± 49.5	F_1,28_ = 3.51	0.07
	Late-lactation	578 ± 42.0	606 ± 47.1	F_1,25_ = 2.49	0.13
BCS	Pre-weaning	4.1 ± 0.3	4.1 ± 0.2	F_1,28_ = 0.05	0.81
	Mid-lactation	4.1 ± 0.4	4.2 ± 0.4	F_1,28_ = 0.40	0.53
	Late-lactation	4.4 ± 0.5	4.5 ± 0.4	F_1,26_ = 0.33	0.57

^1^ Mastitis risk was analysed using chi-square, and LW and BCS were analysed using LMM.

**Table 2 animals-13-02571-t002:** Milk composition of commercial and calf-suckled dairy cows across one lactation. Mean values ± standard deviations are presented. Different superscripts indicate significant differences between commercial and calf-suckled cows within a lactation stage.

Test	Month	Lactation Stage	Fat (%)		Protein (%)		FPCM (kg)	
			Commercial	Suckled	Commercial	Suckled	Commercial	Suckled
1	Aug	Pre-weaning	4.8 ± 0.52 ^a^	4.4 ± 1.40 ^b^	3.5 ± 0.52 ^a^	3.8 ± 0.36 ^b^	26.5 ± 4.81 ^a^	18.1 ± 7.54 ^b^
2	Oct	Mid-Lactation	4.6 ± 1.06	4.4 ± 0.69	3.2 ± 0.40	3.3 ± 0.32	29.4 ± 3.70	27.4 ± 5.04
3	Dec		4.6 ± 1.25	4.3 ± 0.65	3.2 ± 0.44	3.2 ± 0.26	25.5 ± 2.02	25.6 ± 2.71
4	Feb	Late-Lactation	4.5 ± 0.49	4.3 ± 0.69	3.3 ± 0.41	3.3 ± 0.25	22.8 ± 2.00	22.2 ± 3.17
5	Mar		4.7 ± 0.56	4.5 ± 0.64	3.5 ± 0.42	3.4 ± 0.33	22.2 ± 2.53	21.4 ± 2.99
6	May		5.1 ± 0.66	4.8 ± 0.63	3.9 ± 0.46	3.8 ± 0.40	19.2 ± 3.96	18.4 ± 4.12

**Table 3 animals-13-02571-t003:** The effect of treatment on the time spent (hours) ruminating, resting and feeding in commercial and calf-suckled dairy cows during 3 weeks pre- and post-weaning. Different superscripts indicate significant differences between commercial and calf-sucked cows within a lactation stage.

Lactation Stage	Behaviour	Treatment		Test Statistic	*p*-Value
		Commercial	Suckled		
Pre-weaning	Ruminating	515.4 ± 31.04 ^a^	493.2 ± 22.80 ^b^	F_1,28_ = 5.05	0.03
	Resting	285.6 ± 20.37 ^a^	325.6 ± 48.52 ^b^	F_1,28_ = 8.22	0.008
	Feeding	414.7 ± 52.20 ^a^	524.3 ± 43.33 ^b^	F_1,28_ = 39.50	<0.001
Post-weaning	Ruminating	523.6 ± 32.22	513.5 ± 22.84	F_1,28_ = 1.01	0.32
	Resting	271.2 ± 31.00	292.9 ± 44.27	F_1,28_ = 2.34	0.14
	Feeding	416.3 ± 60.34	397.4 ± 66.75	F_1,28_ = 0.66	0.43

## Data Availability

Data are available on reasonable request from the authors.

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
