# Peer review of "A Pilot Study on the Feasibility of an Extended Suckling System for Pasture-Based Dairies"

_animals, 2023, doi:10.3390/ani13162571_

Round 1

Reviewer 1 Report

GENERAL COMMENTS: the authors submitted a pioneer pilot study regarding an extended suckling system for pasture-based dairies. As commented in the first paragraph of the Discussion, this is the most extensive and comprehensive study of a potential suckling system for calves raised in a pasture-based system. Other studies performed a trial with few animals and for indoor dairies. Thus, this highlights the importance of this study. The article is very well-written, complete and objective. I did not see any flaw which compromised the scientific soundness. Below are some minor checks which help improve the final version.

MATERIAL AND METHODS: 

- Figure 1: which are AM and PM, regarding the paddocks? I do not know the meaning of this abbreviation. Please insert the purpose in extensive in the captions.

- Statistical analysis: the description of the longitudinal model is helpful for the scientific audience. 

RESULTS:

- The authors did not insert all the results in tables. The example is shown in the first paragraph of item 3.2 (line 387). Please insert these results in the tables. It isn't easy to compare results when they are stated throughout the text.

Data availability: I don't see reasons to request data from the authors. I'd like to suggest (only suggestion - not mandatory) to insert data in a public repository.

Author Response

We thank the reviewers for dedicating their time and providing expertise while reviewing our manuscript. We have carefully considered the reviewer comments as detailed in our responses that are outlined below. The line numbers in this document correspond to the revised manuscript which records the edits made with tracked changes.

Reviewer 1

Figure 1: which are AM and PM, regarding the paddocks? I do not know the meaning of this abbreviation. Please insert the purpose in extensive in the captions.

We apologise for our inconsistency with labelling of the night (PM) and day (AM) paddocks. This has been rectified in Figure 1 and its associated caption (L226- L232).

Statistical analysis: the description of the longitudinal model is helpful for the scientific audience.

Thank you for this positive feedback!

The authors did not insert all the results in tables. The example is shown in the first paragraph of item 3.2 (line 387). Please insert these results in the tables. It isn’t easy to compare results when they are stated throughout the text.

We have now inserted the mastitis data in Table 2, as suggested by the reviewer.

Data availability: I don’t see reasons to request data from the authors. I’d like to suggest (only suggestion - not mandatory) to insert data in a public repository.

Thank you for your suggestion regarding data availability. We understand the importance of research transparency and data sharing with the scientific community and are fundamentally supportive of the suggested data sharing approach. However, given the novel nature of this study and that this research forms part of a PhD program, we would feel more comfortable waiting until the thesis is submitted before making the data available.

Reviewer 2 Report

This pilot study compares an alternative cow-calf rearing system compatible with pasture access to a conventional system. Much research is needed in this area since there is a growing interest towards alternative rearing systems for dairy calves. Early adopters of cow-calf suckling systems will likely be organic producers, who are required to provide their animals with pasture access. I read this manuscript with great interest. The introduction is complete and concise; hypotheses are clearly stated. The study is well designed and the methodology clearly explained. My only concern about the results is the fact that ADG should, in my opinion, take birth weight into account. The discussion is well structured. I find this manuscript suitable for publication with relatively minor edits.

Specific comments:

Line 54: I think it should read: …in relation to farm animal management.

Line 60: (review: ([6]). It looks like you need to remove one extra bracket before [6].

Line 76: six male cow-calf pairs sounds a bit strange to me. The cow is definitely not male.

Lines 118-119: It is too late to do anything about it, but I am curious to know why you did not move an extra pair to the control group (you already moved 2 pairs) to balance the treatment groups?

Lines 172-173: provide information about rubber ball (size, product info).

Line 178: how was the fresh allocation of pasture estimated? Is the 15.5 kg fresh? Could it be reported in DM?

Line 179: I am not sure that I understand what you mean by ‘fed in the bale’. Was concentrated mixed with silage into a partial mixed ration (PMR), or just sprinkled on top? (also unclear in line 242).

Figure 1: Nice figure. Is the PM paddock and the night paddock the same thing? I find this terminology a bit confusing; why not use day (as in line 249) and night paddocks?

Line 233: …when separated, grain was also offered… (no need to capitalize grain).

Line 239: is 30 minutes sufficient to refill udder to a significant extent before reunion? Was this based on previous research?

Lines 319-320: suckling calves were weighed before joining their dam, so would have been without milk since the day before. Commercial calves had free access to milk during the night. This possible bias should be discussed.

Lines 377-379: you report averages for each treatment, but the association between treatments and averages is not clear when averages are comparable (one needs to look at the figure to find out).

Line 391: …was not significantly different…

Section 3.4 Given the large difference in calf weights between treatments at birth, I would analyze average daily gain with birth weight as a covariable in the model.

Line 422: I am not sure what ‘despite this’ refers to. The previous paragraph refers to the heavier birth weight of suckled calves. There is no mention in the results section of the lower ADG in suckling calves between week 0 and week 3 (unless I missed it). It is in the discussion.

Line 541: I suggest:…to break the nutritional bond before physical separation.

Lines 557-561: the explanation for the lower growth rate of suckled calves during the first 3 weeks in terms of thermoregulation makes sense (very young animals are possibly more susceptible to lower temperatures). I find the ‘activity explanation’ less convincing.  In theory, their increased activity should have been maintained beyond week 3. Were they better able to compensate for this as they got older? Perhaps…

No major issues detected. The manuscript read very well.

Author Response

We thank the reviewers for dedicating their time and providing expertise while reviewing our manuscript. We have carefully considered the reviewer comments as detailed in our responses that are outlined below. The line numbers in this document correspond to the revised manuscript which records the edits made with tracked changes.

Reviewer 2 specific comments:
Line 54: I think it should read: …in relation to farm animal management.
Thanks, we have accepted these suggestions (L54).

Line 60: (review: ([6]). It looks like you need to remove one extra bracket before [6].
Accepted (L65).

Line 76: six male cow-calf pairs sounds a bit strange to me. The cow is definitely not male.
Yes, you are right! We apologise for the oversight. The text has been revised (L81-82).

Lines 118-119: It is too late to do anything about it, but I am curious to know why you did not move an extra pair to the control group (you already moved 2 pairs) to balance the treatment groups?
Our studied cows underwent behavioural testing six months prior to calving so that we could observe relationships between maternal behaviour and temperamental characteristics (subject of a future publication). The final cow in our experiment to calve was assigned to the control treatment but gave birth to a male calf and so was excluded from the trial. By this stage the 16 cows and calves in the suckling systems had already bonded.

Lines 172-173: provide information about rubber ball (size, product info).
We have now included additional information about the ball in the revised manuscript (L177-179).

Line 178: how was the fresh allocation of pasture estimated? Is the 15.5 kg fresh? Could it be reported in DM?
Thanks for this comment. You are correct, the fresh pasture allocation was estimated based on DM. We have included this clarification in the manuscript (L184).

Line 179: I am not sure that I understand what you mean by ‘fed in the bale’. Was concentrated mixed with silage into a partial mixed ration (PMR), or just sprinkled on top? (also unclear in line 242).
Yes, you are right this isn’t clear. The cows were not fed a PMR, the silage was distributed into a trough and the concentrate sprinkled on top. This has been clarified at L184-185 and L248-252.

Figure 1: Nice figure. Is the PM paddock and the night paddock the same thing? I find this terminology a bit confusing; why not use day (as in line 249) and night paddocks?
Reviewer 1 also raised that the use of AM/PM was confusing. We have clarified these terms in the Figure and associated caption.

Line 233: …when separated, grain was also offered… (no need to capitalize grain).
Good suggestion – this change has been made.

Line 239: is 30 minutes sufficient to refill udder to a significant extent before reunion? Was this based on previous research?
Great question. When workshopping this problem, we were unable to find research to guide a decision. However, we did find a fact sheet on nurse cows (cows that suckle multiple calves – sometimes known as foster cows), which was published in New Zealand. Unfortunately, this no longer seems to be available online. We also discussed this issue with a local farmer that runs his own foster-cow calf rearing system. While 30 minutes may not have substantially filled the udder, it seemed to rest the teats enough to prevent further damage.

Lines 319-320: suckling calves were weighed before joining their dam, so would have been without milk since the day before. Commercial calves had free access to milk during the night. This possible bias should be discussed.
Yes you are right. We acknowledge this potential bias. The difference in feeding schedules could have influenced the weight measurements and should be considered when interpreting the results. We have included this at lines 618-623.

Lines 377-379: you report averages for each treatment, but the association between treatments and averages is not clear when averages are comparable (one needs to look at the figure to find out).
We thank reviewer 1 for this comment. This has been rectified at L395.

Line 391: …was not significantly different…
This change has been made.

Section 3.4 Given the large difference in calf weights between treatments at birth, I would analyze average daily gain with birth weight as a covariable in the model.
We agree with the reviewer. We have performed this analysis and made associated changes to the text in the methods L333-336 and 384-386, and results at L4436-441. Including birth weight as a covariate did not change our results or interpretation.

Line 422: I am not sure what ‘despite this’ refers to. The previous paragraph refers to the heavier birth weight of suckled calves. There is no mention in the results section of the lower ADG in suckling calves between week 0 and week 3 (unless I missed it). It is in the discussion.
The analysis of ADG between weeks 0 and 3 are now included (see previous comment) and the words “despite this” removed (L4436-442).

Line 541: I suggest:…to break the nutritional bond before physical separation.
This change has been made.

Lines 557-561: the explanation for the lower growth rate of suckled calves during the first 3 weeks in terms of thermoregulation makes sense (very young animals are possibly more susceptible to lower temperatures). I find the ‘activity explanation’ less convincing. In theory, their increased activity should have been maintained beyond week 3. Were they better able to compensate for this as they got older? Perhaps…
We accept this comment. We have modified our text to better emphasise thermoregulation as a more likely reason why suckled calves grew less from weeks 0-3, and offered increased activity as a possible but less likely explanation (L586-594).

Reviewer 3 Report

Dear authors, thank you for this nice paper on your interesting research.

I have one overall comment: you are stating several times (in the short summary and in the introduction part) that efforts to develop cow-calf suckling systems are due to societal concerns, societal demands, public expectations, etc. But I think it would be worth mentioning that also many farmers themselves have those concerns about animal welfare and aims to develop / ameliorate it. They don't just look for new ways because of the society, but also because of their own observations and their commitment for animal welfare. One recent study showing that aspect is this one:

Please add this aspect.

Specific remarks:

Line 26: replace "commercially" by "as usual"  (because also the other group was actually managed commercially).

Line 29: replace "twice-a-day" by "twice a day"

Line 31: replace "higher fat" by "lower fat" and replace "lower protein" by "higher protein"

Line 54: add "to" between "relation" and "farm" Here you could add a statement on farmers' engagement and interest in animal welfare

Line 111: could you also balance for lactation number or was this not necessary? Please sho the lactation numbers (averages) of both groups. Please say here which breed(s) you worked with.

Line 112: add "for female calves" after "sexed semen"

Line 116: replace "commercially" by "as usual", in the following called "commercial"

Line 118: what are the breeds exactly? (Friesian x Holstein or Jersey?)

Line 165: add "milled" or "flaked" before "grains" (however it was...)

Line 170: Did the commercial calves get any roughage? (following animal protection laws in several European countries roughage has to be fed to calves; if it this is not the case in Australia and calves did not get any roughage, it might be good to mention that here.)

Lines 184/185: please spell out "AM" and "PM"; what does it mean? does it mean for the time slots before and after noon? Please explain shortly

Line 233: replace "Grain" by "grain"

Line 234: replace "once-a-day" by "once a day"

Line 258: Did the calves get roughage in their pen?

Line 287: in the second brackets: replace "week" by "weeks"

Line 314: please explain the BCS Scoring system a little closer: How many scores are used with which intervals?

Line 336: insert a point after "herd". add "of both treatment groups" after "Each cow" (if that is true; otherwise explain how / when you observed behaviour in the commercial cow group)

Chapter 2.9: please split this chapter into "analysis of cow data" and "analysis of calf data"

Lines 347/348: this sentence can stand at the beginning or at the end of the cow-data-analysis chapter.

Line 349: replqce "analysis" by "models" (because post-weaning milk production data were not removed from analysis since you show analysed data below, but they were removed from the models, as i understand it)

Line 355: replace "statistically" by "statistical"

Line 367: replace "calf" by "calves"; replace "was" by "were"

Line 368: replace "gain" by "gains"

Line 370: delete "the" after "on" and delete "the" after "at"

Lines 388 - 391: are you sure that there is no significant difference? It seems to me that suckled cows really had a lower risk. Maybe you could calculate that with cross tables and chi square test?

Line 391: replace "significant2 by "significantly"

Line 400: here you say "sucked cows", but in the table you say "suckled". "Sucked would be actually the right expression and it should be consistent.

Line 404: add "in both treatment groups" after "lactation"

Line 412: add "by chance" before "5 kg heavier"

Line 432: is this total meant over 11 weeks? Please add the time span

Line 451: were there no parasitology tests carried out in commercial calves? It should have been done, because eimeria can also be found in housed animals. If it was not done, this should be mentioned anyway.

Line 476: Add: "Besides that they consumed less grains than commercial calves."

Line 484: replace "Offset" by "compensated"

Line 485: another compensation is the smaller amount of grains that had to be fed to the suckling calves. Don't you want to add that? It is of course a benefit from a "feed-no-food"-point of view, but also economically, depending on the country / region you look at.

Line 499: Nicolao et al., 2022 found the same differences in fat and protein content and they also discussed that quite well. I think you should add that reference here (you have already cited that paper)

Line 507-510. Please explain that closer (have a look at Nicolao et al.)

Line 550: add "by chance" before "5 kg"

Line560. replace "than" by "compared to"

Lines 571-572: Nicolao et al., 2022 also showed that. It would be good to refer to their paper here.

Line 481: could there also be a difference in feeding grains between the two studies?

Chapter 5.: This is mainly a short summary (except the last sentence). Please rewrite that chapter and show your conclusions: what would you recommend referring to your results? Does cow-calf-contact make sense within pasture based systems? from an animal welfare and an economic point of view?

Line 610: delete "Please add"

I made some small remarks (above)
